# Urban–Rural Comparison of the Association between Unsupportive Relationships, Perceived Stress, Authentic Self-Presentation, and Loneliness among Young Adults in Taiwan

**DOI:** 10.3390/ijerph19148808

**Published:** 2022-07-20

**Authors:** Yuting Sun, Chaoyun Liang

**Affiliations:** 1School of Public Administration, Hohai University, Nanjing 210098, China; 20220019@hhu.edu.cn; 2Department of Bio-Industry Communication and Development, National Taiwan University, No. 1, Sec. 4, Roosevelt Road, Taipei 10617, Taiwan

**Keywords:** authentic self-presentation, loneliness, perceived stress, unsupportive relationship, urban–rural comparison, young adults

## Abstract

This study examined and compared how unsupportive relationships, perceived stress, and authentic self-presentation influence loneliness and what differences exist between these relationships across urban and rural young adults in Taiwan. In total, 356 young adults (188 urban and 168 rural) were investigated, and partial least squares structural equation modelling was used in this study. The results indicated that authentic self-presentation directly and negatively affects loneliness in the urban group, but only indirectly affects loneliness in the rural group through unsupportive relationships and perceived stress. Unsupportive relationships and perceived stress in both the urban and rural groups positively affect their loneliness. In addition, multiple group analysis revealed that significant differences only existed between the effects of authentic self-presentation on unsupportive relationships between urban and rural young adults.

## 1. Introduction

Loneliness and health are clearly interrelated [1]. When loneliness is both severe and prolonged, it harms an individual’s mental health [2]. Loneliness involves the individual perception of social relationships being of unsatisfactory quality or failing to meet one’s expectations [3]. Among several factors that predict loneliness, an unsatisfying interpersonal relationship has a key effect on loneliness [4]. Empirical evidence has also demonstrated that both stress and social support are associated with loneliness [5]. In fact, interpersonal support helps individuals maintain or regain strength, particularly when they are under stress or encountering stressful life events, and thus, it decreases the potentially detrimental consequences of stress [6]. Furthermore, low social desirability of a person, evident in changes in social networks and social mobility, induces individual loneliness by limiting opportunities for social interaction [2].

Most people experience various degrees of loneliness at specific points in their lives. Loneliness is not restricted to old age; it can be salient during all developmental stages, especially late adolescence and early adulthood, which have been the focus of recent studies [5,7]. Loneliness among young adults has been a well-studied topic since the early 90s [8]. Early adulthood is a critical transition phase in the individual life span [9]. Young adults in their 20s are at a turning point in life, where they experience loneliness by leaving their hometown and parents to live an independent life and by losing touch with the people who composed their previous social network. This may trigger a strong desire to create new relationships [5,10]. In addition, after graduating from college, young adults either choose to continue their studies or to enter professional life. Consequently, numerous people at approximately the age of 30 may experience a crisis in which they question life’s meaning and previous choices [11]. Therefore, at this transient stage of the life span, it is relevant to focus on the mental health of the age group between 20 and 30.

Prior research has mainly focused on identifying the factors that predict loneliness, and less attention has been given to the associations between the identified factors. Several studies, particularly those conducted in the past 10 years, have investigated the association between interpersonal relationships (social support) and strategic/protective self-presentation (impression management)—which serve as measures of social desirability or self-control—and loneliness, [5,8,12,13,14]. However, few studies have focused on perceived unsupportive interpersonal relationships and authentic self-presentation; moreover, the relationship between loneliness and cultural values is relatively unclear. This is particularly the case for the difference between urban and rural contexts. To bridge this research gap, the current study aims to (1) examine how unsupportive relationships, perceived stress, and authentic self-presentation affect loneliness among young adults and (2) compare the effects across urban and rural regions in Taiwan. This study builds a theoretical foundation and employs partial least squares structural equation modelling (PLS-SEM), including a number of recently developed advanced analysis techniques [15], to address the aforementioned issues. The research outcomes are expected to provide insights into interventions for emotional problems of urban and rural young adults.

## 2. Literature Review

### 2.1. Association of Unsupportive Relationships, Perceived Stress, and Loneliness

Social support is a type of social resource that is gained through an individual’s interpersonal relationships [13]. Interpersonal relationships, such as friendships and other social bonds, play a vital role in an individual’s life and personal development [14]. The quality of such relationships depends on the responsiveness of one’s family members, peers, friends, and partner; this quality is evaluated on the basis of whether the people important to an individual understand and support the individual [16]. The lack of responsiveness and a supportive network may influence the quality of interpersonal relationships, which thwarts an individual’s sense of security, satisfaction, closeness, and commitment in a relationship [16].

Loneliness is a subjective experience and an aspect of perceived well-being [17]. It is characterised by unpleasant feelings that result from subjective deficiencies in an individual’s interpersonal relationships [18] or a lower level of satisfaction than that desired by the individual [8]. A supportive relationship contributes to mental well-being and life satisfaction [19]. In particular, emotional social support can be viewed as high-quality rapport in which individuals feel respected and understood. Conversely, poor relationship quality can generate stress and undermine health and well-being [20,21]. Support from various close relationships, such as parents, partners, and siblings, mitigates loneliness to various degrees during distinct life stages [22]. A previous study reported social network conditions that are associated with loneliness among college students [8]. Whether individuals can maintain harmonious interpersonal relationships is a crucial question in loneliness research concerning young adults.

Perceived stress refers to the degree to which an individual views his or her life as stressful; it differs from the actual feeling and symptoms of stress itself [23]. Self-conceptions [24], social capital [25], and social networks [26] are viewed as possible antecedents of stress. An unsupportive relationship is conceptualised as one form of chronic stress that engenders major depression or depressive symptomatology [27]. Previous studies have indicated that people in unsupportive relationships are less resilient to stress than those with more supportive relationships [28,29]. When people fall into troubled relationships, stress and depression are often experienced simultaneously [30]. People with unsatisfying interpersonal relationships receive little social support to cope with various stressors in life [26]. When young adults face increasing stress, they may experience depressive mood, dissatisfaction, a sense of failure, irritability, and other symptoms associated with depression [31].

University life requires students to become independent, which may induce a reformation and possible disintegration of crucial social support systems [32], including their previous social relationships. This pressure and reformation of social relationships may trigger feelings of stress, loneliness, and learning burnout. In general, the development of these stressors leads to further negative effects on students, such as poor mental and physical health. Therefore, this study proposed the following three hypotheses:

**H1.** 
*Unsupportive relationships positively affect loneliness.*


**H2.** 
*Unsupportive relationships positively affect perceived stress.*


**H3.** 
*Perceived stress positively affects loneliness.*


### 2.2. Authentic Self-Presentation Related to Unsupportive Relationships, Perceived Stress, and Loneliness

Social psychology distinguishes two approaches to self-presentation: strategic and authentic [33]. Strategic/protective self-presentation, also called impression management, is a crucial factor widely used by researchers and practitioners to assess whether individual bias or self-report are conducted in a self-favouring manner [13,33,34,35]. For strategic self-presentation, an individual uses conscious and deliberate efforts to shape others’ impressions in order to gain outcomes such as power, sympathy, and influence. Thus, some people pay substantial attention to the maintenance of a positive public image to avoid negative evaluation by the public [35,36]. By contrast, authentic self-presentation is not aimed at impression management but rather at the verification of the crucial aspects of one’s self-concept [33]. Individuals with a low level of public self-consciousness but a high level of private self-consciousness do not exhibit these strategic patterns [37].

A previous study indicated that strategic self-presentation, peer relationships, and perceived social anxiety exert unique effects on loneliness. In particular, strategic self-presentation is highly associated with loneliness [38]. Lonely people tend to use strategic self-presentation to avoid being perceived negatively, and they are less willing to reveal their authentic selves [39]. A high level of strategic self-presentation leads to substantial social anxiety [40]. People with strategic self-presentation also invest considerable effort in this process, which may be laborious and stressful [13]. Individuals who manage their social impression are reluctant to show their hardships and struggles to others; hence, such individuals are less likely to receive support from their friends and colleagues because colleagues perceive little need to provide support [13]. Therefore, strategic self-presentation is associated with an unsupportive interpersonal relationship and a lower level of social connectedness.

Strategic self-presentation is risky; even if an individual is successful at strategic self-presentation in the short term, the durability of an ideal outcome is a matter of concern [41]. This is because strategic self-presentation may backfire and be detrimental; for example, individuals who employ strategic self-presentation to be considered friendly or capable are often perceived as flatterers or boasters, respectively. By contrast, authentic self-presentation may persist in the long term. However, recent studies have focused more on the effects of strategic/protective self-presentation and less on those of authentic self-presentation. As an adjustment approach, authentic self-presentation is related to a high-motivation/competence model associated with low stress, friendliness, good interpersonal relationships, and enhanced well-being [34].

Self-presentation has been reported to vary on the basis of age, sex, culture, and social environment [42]. In particular, college students, from early adolescence through adulthood, may be eager to establish interpersonal relationships and reduce the feeling of loneliness owing to changes in their lives, careers, and environments. Furthermore, in Chinese culture, strategic self-presentation is based on the ability to control oneself and maintain harmony with the social context [43]. Taiwan’s cultural uniqueness and diversity result from the influence of traditional Chinese and Western cultures. Taiwanese society has been influenced by Confucian collective culture, which gives priority to harmony and emphasises the high need for social identity. However, in line with the ‘modern values’ absorbed from Western culture, Taiwanese society is gradually transforming into one that places more emphasis on individuality and autonomy but less emphasis on order and obedience [44]. Accordingly, investigating the association between authentic self-presentation, unsupportive relationships, perceived stress, and loneliness is imperative. Consequently, this study proposed three hypotheses as follows:

**H4.** 
*Authentic self-presentation*
*negatively affects unsupportive relationships.*


**H5.** 
*Authentic self-presentation*
*negatively affects loneliness.*


**H6.** 
*Authentic self-presentatio*
*n negatively affects perceived stress.*


### 2.3. Comparison of Loneliness Patterns between Urban and Rural Young Adults

The literature demonstrates the fluidity and dynamism of cultural transmission as well as the significance of rural–urban migration through the migrants’ individual agency and ‘self-realisation’ [45]. Durkheim [46] replaced the concept of ‘social integration’ with that of ‘social solidarity’ and proposed ‘mechanical solidarity and organic solidarity’ based on the social division of labour and differences in the collective consciousness. In a ‘traditional’ rural society, ‘mechanical solidarity’ emphasises a sense of identity where close, personal relationships predominate moral and social connections. In a ‘modern’ urban society, ‘organic solidarity’ refers to the functional interdependence that originates from the division of labour based on highly differentiated, specialised, and impersonal economic roles.

Taiwanese society has transformed into an industrial society over the past three decades. Owing to the rapid socioeconomic changes in Taiwan, enhanced population mobility disintegrated the relatively stable urban–rural dual structure, disrupted the social division of labour, and increased population aggregation; population mobility creates an urban space and promotes urban development. Social integration can be reshaped through new social networks and lifestyle patterns based on different groups in urban and rural areas in Taiwan. The evolution of social dynamics has resulted in a developmental context where Taiwanese youths are increasingly engaging in nonfamilial social interactions [47].

Previous studies have demonstrated extremely high loneliness among rural adolescents because of limited opportunities for social interactions [48], especially for rural male adolescents [49]. Prior research indicated that rural adolescents’ social loneliness is significantly higher than their emotional loneliness [49]; the social loneliness of rural adolescents is significantly higher than that of urban adolescents [48], and the emotional loneliness of rural adolescents is lower than that of urban adolescents [50]. This indicated that high-quality friendships are more crucial than the number of friends one has among rural adolescents [48,50]. However, most studies have focused more on the urban–rural comparison of loneliness in adolescents; little attention has been given to the transition stage. Moreover, although studies have reported that loneliness is associated with family relationships [51] and peer attachment [52], the relationship between loneliness and cultural values remains somewhat unclear. Accordingly, the following seventh hypothesis was proposed:

**H7.** 
*There are significant differences in the patterns by which loneliness influences urban and rural young adults.*


## 3. Method

### 3.1. Data Description

The data were collected from the seventh (2017) Taiwan Social Change Survey —Social Networks and Social Resources. The data collected from a total of 358 townships/districts in Taiwan were classified into seven categories: metropolitan cities, urban cities, rising cities, traditional industrial townships, rural townships, aged townships, and remote townships. This classification was performed on the basis of the following variables: ‘rural employment as a proportion of the total employment’, ‘nonrural employment as a proportion of the total employment’, ‘professionals and executives as a proportion of the total employment’, ‘population aged between 15 and 64 years as a proportion of the total population’, ‘population aged ≥ 65 years as a proportion of the total population’, ‘population with university education and above as a proportion of the total population’, ‘population density’, and ‘population growth in the last 5 years’ [53]. A multistage cluster sampling process was performed using the probability proportional to size method to randomly select individuals aged > 18 years old from Taiwan’s population data for the survey [54]. The original sample size for the survey was 1955, and the selected sample had the same general characteristics as the entire population of Taiwan. Each respondent received a gift card after completing the survey questionnaire.

### 3.2. Participants

Our sample consisted of 356 young adults (188 males and 168 females) aged from 19 to 29. The mean age was 23.91 years for males and 21.91 years for females. One of the items on the self-report questionnaire is outlined as follows: ‘Which of the following applies to your current residential place? (A) metropolis, (B) suburban, (C) township, (D) rural area, or (E) independent farmhouse’. This item helped us identify the respondents’ residential location in 2017. On the basis of the responses recorded, we further divided the study sample into two groups: urban and rural. Although the respondents recognised that they lived in a township, they preferred to select option D (‘rural area’). To avoid statistical errors, the respondents were distributed relatively evenly between the urban and rural groups. Respondents who selected option A (‘metropolis’) and option B (‘suburban’) were classified into the urban group (*n* = 188), and the remaining respondents were classified into the rural group (*n* = 168). The sociodemographic characteristics of the samples are presented in Table 1.

### 3.3. Measurement

As a negative dimension of interpersonal relationship quality, an unsupportive relationship was assessed in terms of a negative relationship with the spouse, from the negative aspect of a nine-item marital relationship quality scale [55]. The items on the aforementioned scale are outlined as follows: ‘How often do you feel that your family, relatives, and friends are demanding?’, ‘In the past 4 weeks, how often have you felt nervous because of the important people in your life?’, and ‘How often do you feel burdened by your interpersonal relationships with your relatives and friends?’. Response categories were 1 = never, 2 = rarely, 3 = sometimes, 4 = often, and 5 = very often. Higher scores indicated higher levels of unsupportive interpersonal relationships.

Perceived stress was measured using the 20-item Perceived Stress Scale, which describes life stress in terms of feeling in control [23]. The two items on this scale were ‘How often do you get upset?’ and ‘How often do you feel anxious and worried?’. Response categories were 1 = never, 2 = rarely, 3 = sometimes, 4 = often, and 5 = very often. Total scores summed across the three items defined each participant’s level of perceived stress. Higher scores indicated higher levels of perceived stress.

Authentic self-presentation, as a component of social desirability, was evaluated using a short form of the 16-item Balanced Inventory of Desirable Responding scale [56]. This subscale helped evaluate the tendency of the participants to reveal their dark sides and provide an honest self-description. The subscale items were as follows: ‘I have occasionally taken advantage of others’ and ‘I like to gossip about others’. The items were rated as follows: 1 (‘never’), 2 (‘rarely’), 3 (‘sometimes’), and 4 (‘often’). Higher scores indicated higher levels of authentic self-presentation.

Loneliness was evaluated on the basis of four items retrieved from the eight-item short-form of the UCLA Loneliness Scale [57]. The scale items were as follows: ‘How often do you feel lonely?’, ‘How often do you feel that you lack companionship?’, and ‘How often do you feel isolated from others?’. The items were rated using a 5-point scale, 1 = never, 2 = rarely, 3 = sometimes, 4 = often, and 5 = very often. The mean scores were calculated; higher scores indicated higher levels of loneliness.

### 3.4. Procedure

Descriptive statistical analysis was performed to calculate means, standard deviations, and factor loadings. In addition, the reliability and validity of each variable were assessed. To determine any possible differences between the urban and rural groups, Student’s *t*-test was used to calculate the means of major variables. The PLS-SEM method [58] was used in this study. PLS is a multivariate analysis approach to estimate path models with latent variables [59], and it supports the performance of multigroup analysis (MGA) while maximising the explained variance in the dependent variables and minimising the residual variance [60,61]. Bootstrapping in PLS offers *t*-test statistics that confirm the statistical significance of the predictors. Calculation of all PLS results was performed with SmartPLS 3.2.8 statistical software (www.smartpls.com), which was developed by SmartPLS GmbH Company in Boenningstedt, Germany [62]. The reporting of results of PLS-SEM has been done in two steps; one is the measurement model with reliability and validity of item analysis, and the other is a structural model [63].

## 4. Results

### 4.1. Descriptive Analysis and Measurement Model Assessment

The independent samples *t*-test results revealed that, except for loneliness, urban and rural young adults exhibited significant differences in their unsupportive relationships, perceived stress, and authentic self-presentation. On average, the urban group had higher unsupportive relationships and perceived stress compared with the rural group, whereas the urban group had lower authentic self-presentation compared with the rural group. These findings imply that urban young adults in Taiwan experience more stress and negative emotions than rural young adults. The Cronbach’s alpha of four scales achieved 0.6, and the internal reliability of this study was confirmed (Table 2).

A confirmatory factor analysis with maximum likelihood estimation was used to confirm the factor structures of the four subscales. The analyses yielded a good fit (*χ*^2^ = 403.26, d_ULS = 0.38, d_G = 0.23, standardised root mean square residual [SRMR] = 0.06, NFI = 0.81). The data reliability and validity were examined with respect to the latent variables and their associated items [61,64], including composite reliability (CR), convergent validity (CV), and discriminant validity (DV), which can be evaluated by performing PLS algorithm procedures. The results indicated that the CR coefficients in the PLS path model were higher than 0.7, achieving acceptable levels [61,64]. The results also indicated that all average variance extracted (AVE) values were higher than 0.5 [65]. Therefore, CV was warranted. In addition, the heterotrait–monotrait ratio of correlations (HTMT) was used in this study to examine DV because HTMT has recently been established as a superior criterion compared with the more conventional methods such as the Fornell–Larcker criterion [66,67]. The results (Table 3) indicated that HTMT criterion values were below 1.0 [60], confirming DV.

The *R*^2^ value calculated is indicative of a model’s explanatory power [68]. The values of this model were 0.117 for unsupportive relationships, 0.349 for perceived stress, and 0.379 for loneliness. Except for unsupportive relationships, the *R*^2^ values of other variables were relatively high [68]. Regarding the effect size *f*^2^ for the structural model, unsupportive relationships had a strong effect size of 0.311 on perceived stress and a weak effect size of 0.049 on loneliness. Perceived stress had a medium effect size of 0.128 on loneliness. Authentic self-presentation had a weak effect size of 0.058 on loneliness, a weak effect of 0.066 on perceived stress, and a medium effect size of 0.132 on unsupportive relationships. Finally, we tested the model fit through SRMR. The SRMR in this model was 0.083, achieving the criterion of 0.08 [60].

### 4.2. Assessment of Structural Model (Hypotheses Testing)

The results of structural model assessment and hypotheses testing (Table 4 and Figure 1) yielded path coefficients with significant levels at a confidence level of 95% [69]. Accordingly, the positive effect of unsupportive relationships of rural young adults on their loneliness was significant, but that of the urban group was not. The unsupportive relationships of both groups positively affected their perceived stress. The perceived stress of both groups also positively affected their loneliness. In addition, authentic self-presentation of both groups negatively affected their unsupportive relationship. Authentic self-presentation of both groups also negatively affected their perceived stress. Furthermore, the negative effect of authentic self-presentation of urban young adults on their loneliness was significant, but that of the rural group was not. Therefore, for urban young adults, H2, H3, H4, H5, and H6 were supported; for rural young adults, H1, H2, H3, H4, and H6 were supported and H7 was supported.

The direct and indirect effects resulting from all the latent predictor variables on loneliness are reported in Table 5. With respect to the urban group, perceived stress had the strongest direct effect on loneliness, followed by authentic self-presentation and unsupportive relationships (0.403, −0.275, and 0.150, respectively). The indirect effect of unsupportive relationships on loneliness was slightly stronger than that of authentic self-presentation (0.196 and −0.154, respectively). With respect to the rural group, unsupportive relationships had the strongest direct effect on loneliness, followed by perceived stress and authentic self-presentation (0.302, 0.299, and −0.131, respectively). The indirect effect of authentic self-presentation on loneliness was stronger than that of unsupportive relationships (−0.267 and 0.135, respectively).

### 4.3. MGA

To test the path differences between urban and rural young adults, a multigroup approach was used. The PLS-MGA method is a nonparametric significance test for difference in group-specific results based on PLS-SEM bootstrapping results [70]. Significant differences were exhibited using both Henseler’s MGA and the permutation test with a *p*-value higher than 0.95 and lower than 0.05 [71]. As reported in Table 6, this analysis revealed that only the effects of authentic self-presentation on unsupportive relationships were significantly different between urban and rural young adults. Both methods confirmed significant and nonsignificant results, which provided multimethod validation, thus increasing the credibility of the current study.

## 5. Discussion

### 5.1. Determinants Pattern in Urban and Rural Young Groups

The results indicated that two major factors affecting loneliness differently in urban and rural young adults were the direct and indirect effect of authentic self-presentation. First, authentic self-presentation directly led to loneliness in the urban group, indicating that the inner states of urban young adults are directly affected by their external self-image. This concurs with prior research [12,34] and implies that individuals with low authentic self-presentation might be sensitive and more likely to perceive high stress and unsupportive relationships. By contrast, rural individuals reflect differently [72,73]. Young adults living in rural areas and having a high level of authentic self-presentation do not experience loneliness instantly but may gradually experience perceived stress and unsupportive relationships. Rural young adults need not excessively enhance their self-image or engage in positive illusions to hide their feeling of loneliness. They are simply content with their current relationship and life quality.

Second, the indirect effects of authentic self-presentation on loneliness are different between urban and rural young adults because of the association of authentic self-presentation with unsupportive relationships and that of unsupportive relationships with loneliness. Regarding the first association, the negative effect of authentic self-presentation on unsupportive relationships was significantly higher in the rural group than in the urban group. This demonstrated that, within the same extent of authentic self-presentation, young adults living in urban areas received a lower level of social support from their network than did those living in rural areas; this finding is consistent with those of previous studies [74,75]. This suggests that authentic self-presentation is more advantageous in maintaining the quality of interpersonal relationships for young adults living in rural areas than for those living in urban areas.

In addition, the effect of unsupportive relationships on loneliness was significant in the rural sample but nonsignificant in the urban sample. Studies have determined that unsupportive relationships increase rural individuals’ loneliness [75,76] and enhance the negative effect of authentic self-presentation on loneliness. Moreover, our aforementioned finding is consistent with those of previous studies reporting that people living in urban Taiwan can access richer resources and they are more concerned about the quantity of their individual social networks than the quality of strong emotional support from friendship [48,50]. As a consequence, the function of relationship quantity plays a crucial role in the loneliness of urban young residents. These results further indicated that when poor relationship quality is perceived to exert stress on one’s life, it causes loneliness (i.e., unsupportive relationship → perceived stress → loneliness).

### 5.2. Differences in Taiwanese Social Context

Nearly one-third of the Taiwanese population resides in the Taipei metropolitan area, but 50 years ago, only 18.4% of the population resided in this area [77]. This indicates that the decrease in rural population size and loss of community vitality narrow down the types of rural interpersonal relationships but enhance the relationship quality and sense of belonging [78]; conversely, the increase in urban population size results in diversified interpersonal relationships but reduced relationship quality and, in turn, a reduced sense of belonging. Young adults who grow up in rural areas often leave their hometowns and parents to lead an independent life in urban areas, thus losing contact with their previous social network; nonetheless, they still have opportunities to create new relationships in the urban areas [5,10]. This environmental change causes a psychological transition and adaption from a family orientation to a self-oriented conception of self among rural young adults. This alteration can entail forming and maintaining friendships and avoiding the unacceptable social outcome of social exclusion [34].

The aforementioned results are closely related to sociocultural context, family structures, and daily practices, which influence self-reported personal values [79]. In this study, higher scores in authentic self-presentation of the rural young adults revealed that collectivist culture and traditional norms are practised more in rural families than in urban families, prioritising harmony and interdependence within a close community as well as contributing to a high need for social approval. Because urban and rural contexts entail dissimilar cultures, the self-image performed by urban/rural dwellers varies in nature and practice [79]. These results indicate that the interpersonal integration observed in urban and rural Taiwan is consistent with Durkheim’s social integration: mechanical solidarity is predominant in urban Taiwan, whereas organic solidarity is evident in rural Taiwan [46].

Furthermore, distinct levels of negative feelings existed among Taiwanese urban and rural young adults in this study. However, the MGA of the two models demonstrated that differences among the five paths were nonsignificant. It further revealed that the disparity between urban and rural areas may be the result of a cumulative process of development [80]. This indicated that, in the future, as suburban areas are integrated into urban ones, young people living in suburban areas might feel the same degree of unsupportive relationships, perceived stress, and loneliness experienced by the current urban young adults in this sample. Another observation in terms of developmental stage is that self-construal directly influences beliefs about interpersonal interaction [81]. The modern Taiwanese (e.g., young adults living in urban areas) actively seek control over their external environment to construct a high belief in the independent self, whereas individuals with traditional values (e.g., young adults living in rural areas) may seek harmony in interpersonal relationships to adapt to the environment [81].

Similarly, previous studies have indicated that urbanisation generally leads to improvement in health and mental health outcomes as a result of a decrease in poverty and improved access to healthcare [82,83]. However, some severe urban problems have flowed from Taiwan’s urbanisation, such as unaffordable housing, lack of job security, life pressure, and urban poverty, creating a stressful environment for young adult development and generating negative emotions and social interactions [84,85,86]. Even though stressors are higher in urban young residents than in rural young residents, the degree of difference is small.

## 6. Conclusions

In this study, we compared the effects of unsupportive relationships, perceived stress, and authentic self-presentation on loneliness among young adults living in urban and rural Taiwan. The findings indicated, first, that unsupportive relationships, perceived stress, and authentic self-presentation of the urban group are significantly higher than those of the rural group. Second, authentic self-presentation has a direct negative effect on loneliness, whereas its indirect influence is mainly through unsupportive relationships for the rural group. Third, the major differences affecting loneliness between urban and rural young adults are the direct and indirect effects of authentic self-presentation. 

The results can further indicate that the determinant patterns in the loneliness of Taiwanese young adults are linked to crucial experiences during their lives (such as college education and beginning a career). Simultaneously, the differences in the determinants of loneliness experienced by young adults living in urban and rural Taiwan are influenced by sociocultural factors and urbanisation levels, including political and economic situations, differences in social and cultural structures, the nature of social problems, and participant ages. In addition, young adults living in rural Taiwan appeared to be more family-oriented, have a higher level of traditional values, and be more concerned about maintaining harmonious relationships than did those living in urban Taiwan; however, those living in urban Taiwan appeared to have a higher level of emotional responsiveness to stress, a higher level of individualistic values, and more intense interpersonal relationships (owing to the fast-paced life trends in urban areas).

## 7. Contributions and Implications

The present study contributed to the existing literature on social issues and human relations by establishing a feasible research framework to study loneliness, validating this framework to become a practical model in the unique Taiwanese context, and identifying loneliness patterns between urban and rural young adults. Therefore, several implications can be drawn from these results. First, stronger interpersonal support, regardless of it being from family, relatives, or friends, is necessary to ensure healthier psychological states for young adults, particularly for those rural young adults far away from home. Second, various hometown associations could be promoted on campus or in corporations that young adults could join. This could not only make the rural group gain a sense of belonging from familiar and supportive relationships but could also help the urban group to relieve stress from intensive academic pressure or burdensome urban work. Finally, patterns between urban and rural young adults in Taiwan are dissimilar. In this line, it is necessary to take into account the fact that government has positively introduced diverse policies related to rural development, urban–rural balance, and facilitated cultural interaction.

## 8. Limitations and Further Development

Some limitations in the current study should be addressed. First, it must be acknowledged that the data used in this study were selected from a national survey rather than by targeting a specific research theme where researchers could control variable scales, which might have led to the overestimation of certain concepts. For this database, the items were collected from various measurement scales; moreover, the numbers of items on the scales were somewhat inconsistent because of the questionnaire’s design. For example, unlike other variables that were measured using 5-point scales in this study, authentic self-presentation was measured using a 4-point scale. Reorganising entire scales to revisit the same samples could generate more objective and accurate results.

Second, to avoid statistical errors, the study sample was further distributed relatively evenly between the urban and rural groups, which might not have represented the general urban–rural population in Taiwan. Third, cross-sectional data were used to examine the hypotheses in this study, but the nature of this type of analysis does not allow for the claim of causal effects. Therefore, the relationships of loneliness determinants identified in this study should be more cautiously interpreted as associations rather than as evidence of the influence. Further studies could therefore introduce a longitudinal research design, as short-term loneliness might develop into prolonged depression if a person is continuously exposed to high stress and a shortage of emotional support from interpersonal relationships.

Finally, although face-to-face surveys increase the level of honesty, collecting data only through self-reporting measures, as used in this study, may have limited sample size and posed a threat to internal validity because of concerns regarding the quality of data solicited by interviewers. Consequently, multiple methods of evaluation should be used to reduce the effect of subjectivity in the future.

## Figures and Tables

**Figure 1 ijerph-19-08808-f001:**
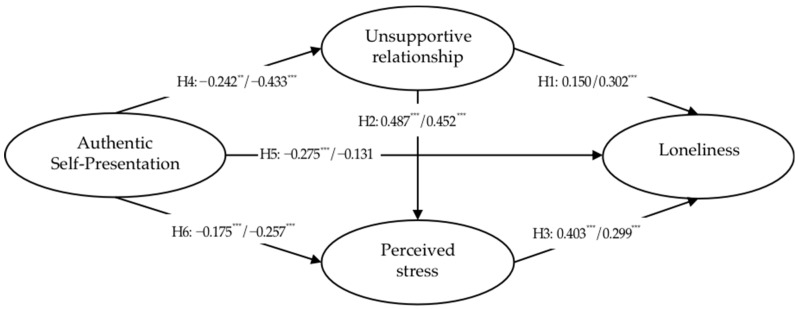
Assessment of structural model (urban *n* = 188, rural *n* = 168). Notes: path coefficients: urban/rural; ** *p* < 0.01, *** *p* < 0.001.

**Table 1 ijerph-19-08808-t001:** Sociodemographic analysis (*n* = 356).

**-**	Urban (*n* = 188)		Rural (*n* = 168)	
	Frequency	Percentage	Frequency	Percentage
Gender	-	-	-	-
Male/Female	95/93	50.5%/49.5%	93/75	55.4%/44.6%
Nation	-	-	-	-
Taiwanese/Others	169/19	89.9%/10.1%	161/7	95.8%/4.2%
Marital status	-	-	-	-
No/Yes	172/16	91.5%/8.5%	154/14	91.7%/8.3%
Living condition (number)	-	-	-	-
Living alone/Live with members ≥ 2	24/164	12.8%/87.2%	15/153	8.9%/91.1%
Education level	-	-	-	-
<College/≥College	78/110	41.5%/58.5%	96/72	57.1%/42.9%
Religion	-	-	-	-
No/Yes	87/101	46.3%/53.7%	59/109	35.1%/64.9%
Individual Income	-	-	-	-
<NTD 10,000/NTD 10,000—NTD 29,999/	45/71/	23.9%/37.8%/	35/67/	20.8%/39.9%/
NTD 30,000—NTD 49,999/>NTD 50,000	57/15	30.3%/8.0%	59/7	35.1%/4.2
Employment situation	-	-	-	-
No/Yes	54/134	28.7%/71.3%	40/128	23.8%/76.2%
Social status	-	-	-	-
Low/Middle/High	20/133/35	10.6%/70.8%/18.6%	27/118/23	16.1%/70.2%/13.7%
Total	188	100.0%	168	100.0%

**Table 2 ijerph-19-08808-t002:** Descriptive analysis and confirmatory factor analysis results (*n =* 356).

Factor or Item	Urban		Rural		*t*	Alpha	FLs	CR	AVE
	*M*	*SD*	*M*	*SD*					
Unsupportive relationship	2.09	0.89	1.84	0.85	3.50 ***	0.72		0.82	0.54
UR1: How often do you feel stressed due to the work done by your family members in your daily life?	2.32	0.98	2.00	0.95			0.79	-	-
UR2: How often do you feel that your family, relatives, and friends are demanding?	2.11	0.92	1.83	0.80			0.78	-	-
UR3: In the past 4 weeks, how often have you felt nervous due to the important people in your life?	2.02	0.79	1.82	0.86			0.66	-	-
UR4: How often do you feel burdened by your interpersonal relationships with relatives and friends?	1.86	0.89	1.71	0.86			0.70	-	-
Perceived stress	2.44	0.64	2.20	1.03	2.56 *	0.85		0.91	0.77
PS1: How often do you get upset?	2.37	0.96	2.14	1.00			0.89	-	-
PS2: How often have you felt that your difficulties have been stacked to such an extent that you could not overcome them?	2.36	0.92	2.21	1.06			0.88	-	-
PS3: How often have you felt anxious and worried?	2.60	1.00	2.26	1.04			0.87	-	-
Authentic self-presentation	3.17	0.73	3.35	0.73	−3.04 **	0.60		0.78	0.55
AS1: There have been occasions when I took advantage of someone.	3.47	0.62	3.63	0.57			0.63	-	-
AS2: There have been times when I was quite jealous of the good fortune of others.	3.13	0.83	3.27	0.81			0.87	-	-
AS3: I like to gossip about others.	2.92	0.73	3.15	0.82			0.70	-	-
Loneliness	1.66	0.82	1.56	0.85	1.33	0.84		0.89	0.68
L1: How often do you feel lonely?	1.79	0.91	1.68	0.97			0.85	-	-
L2: How often do you feel that you lack companionship?	1.73	0.81	1.60	0.93			0.91	-	-
L3: How often do you feel isolated from others?	1.37	0.64	1.33	0.61			0.76	-	-
L4: How often do you feel that your relatives and friends have a better life than you?	1.76	0.94	1.64	0.88			0.75	-	-

FLs refers to factor loadings; CR refers to composite reliability; AVE refers to average variance extracted; Alpha refers to Cronbach’s alpha. * *p* < 0.05, ** *p* < 0.01, *** *p* < 0.001.

**Table 3 ijerph-19-08808-t003:** Discriminant validity (HTMT) (*n* = 356).

Constructs	L	UR	PS	IM
Loneliness (L)				
Unsupportive relationship (UR)	0.602			
Perceived stress (PS)	0.643	0.697		
Authentic self-presentation (AS)	0.546	0.481	0.514	

**Table 4 ijerph-19-08808-t004:** Assessment of structural model.

Hypothesis		Urban (*n* = 188)		Rural (*n* = 168)		Supported
		Path Coefficients	Confidence Intervals (Bias Corrected)	Path Coefficients	Confidence Intervals (Bias Corrected)	
H1	UR → L	0.15	[−0.037; 0.323]	0.3022 ***	[0.1291; 0.4702]	No/Yes
H2	UR → PS	0.487 ***	[0.363; 0.597]	0.4523 ***	[0.3106; 0.5771]	Yes/Yes
H3	PS → L	0.403 ***	[0.227; 0.562]	0.2994 ***	[0.105; 0.4679]	Yes/Yes
H4	AS → UR	−0.242 **	[−0.362; −0.076]	−0.4332 ***	[−0.5307; −0.2824]	Yes/Yes
H5	AS → L	−0.275 ***	[−0.392; −0.132]	−0.1314	[−0.2958; 0.0315]	Yes/No
H6	AS → PS	−0.175 ***	[−0.310; −0.031]	−0.2572 ***	[−0.3766; −0.1319]	Yes/Yes

** *p* < 0.01, *** *p* < 0.001.

**Table 5 ijerph-19-08808-t005:** Effects of latent independent and mediating variables.

Latent Independent and Mediating Variables	Urban (*n* = 188)			Rural (*n* = 168)		
	Direct	Indirect	Total	Direct	Indirect	Total
Authentic self-presentation	−0.275 ***	−0.154 **	−0.429 ***	−0.131	−0.267 ***	−0.398 ***
Unsupportive relationship	0.150	0.196 ***	0.346 ***	0.302 ***	0.135 **	0.437 ***
Perceived stress	0.403 ***	-	0.403 ***	0.299 ***	-	0.299 ***

** *p* < 0.01, *** *p* < 0.001.

**Table 6 ijerph-19-08808-t006:** Results of MGA between urban and rural young adults.

		Urban (*n* = 188)		Rural (*n* = 168)		-	-	-	-
Hypothesis		Path Coefficients	Confidence Intervals (Bias Corrected)	Path Coefficients	Confidence Intervals (Bias Corrected)	Path Coefficient Differences	*p*-Value Henseler’s MGA	*p*-Value Permutation Test	Supported
H1	UR → L	0.150	[−0.037; 0.323]	0.302 ***	[0.129; 0.470]	0.152	0.886	0.227	No/No
H2	UR → PS	0.487 ***	[0.363; 0.597]	0.452 ***	[0.311; 0.577]	0.035	0.347	0.693	No/No
H3	PS → L	0.403 ***	[0.227; 0.562]	0.299 ***	[0.105; 0.468]	0.103	0.187	0.378	No/No
H4	AS → UR	−0.242 **	[−0.362; −0.076]	−0.433 ***	[−0.531; −0.282]	0.192	0.020 *	0.043 *	Yes/Yes
H5	AS → L	−0.275 ***	[−0.392; −0.132]	−0.131	[−0.296; 0.032]	0.143	0.914	0.164	No/No
H6	AS → PS	−0.175 ***	[−0.310; −0.031]	−0.257 ***	[−0.377; −0.132]	0.082	0.191	0.385	No/No

* *p* < 0.05, ** *p* < 0.01, *** *p* < 0.001.

## Data Availability

The data presented in this study are available on request from the corresponding authors.

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
