# Peer review of "Urban–Rural Comparison of the Association between Unsupportive Relationships, Perceived Stress, Authentic Self-Presentation, and Loneliness among Young Adults in Taiwan"

_ijerph, 2022, doi:10.3390/ijerph19148808_

Round 1

Reviewer 1 Report

It is an interesting study. Here are my comments about the manuscript. 

1. Please clarify the clarification of urbane and rural areas in Taiwan. The authors should provide foundations (such as population) to define urbane and rural areas. Is the group divided based on respondents' residential location in 2017 or where respondents grew up? 

2. Data collection method should be more straightforward. It is a little confusing to understand the data collection method. Please explain how the respondents were selected. Are they paid or not?

3. Please check the scales for all the variables under the measurement section. In particular, unlike other variables, impression management was measured with only 4 scales. 

4. Please check the consistency of italic to specific variables or sentences. 

Author Response

It is an interesting study. Here are my comments about the manuscript. 

  1. Please clarify the clarification of urbane and rural areas in Taiwan. The authors should provide foundations (such as population) to define urbane and rural areas. Is the group divided based on respondents' residential location in 2017 or where respondents grew up?

Response:

We have added more details in Method section 3.1 Data description and 3.2 Participants. Please review page 5 (lines 199-200) page 6 (lines 201-229) of the revised document.

  1. Data collection method should be more straightforward. It is a little confusing to understand the data collection method. Please explain how the respondents were selected. Are they paid or not?

Response:

We have written this part. Please review page 5 (lines 199-200) page 6 (lines 201-229) of the revised document.

  1. Please check the scales for all the variables under the measurement section. In particular, unlike other variables, impression management was measured with only 4 scales. 

Response:

This study used data from an existing database for further analysis. Scales of variables are not consistent caused by the format of the original questionnaire. We have addressed this problem in the limitation section. Please refer to page 13 (lines 476-484) of the revised document.

  1. Please check the consistency of italic to specific variables or sentences. 

Response:

We have checked and revised accordingly.

Reviewer 2 Report

This is an interesting study on perceived loneliness between two regions of Taiwan. While the methods and analysis are clearly presented, the front end of the paper needs quite a bit of revision. For one, loneliness among young adults is well studied and began to gain traction in the early 90s. Next, the relationship between interpersonal relationship (support) and impression management and loneliness has also been widely researched, particularly in the last 10 years.  I would like to see this acknowledged in your review of the literature (e.g. Ponzetti (1990), Wang et al. (2020)) as the initial implication is that the variables used in this study are somewhat novel.

Next, the variables need to be more clearly operationalized. Impression management in particular needs to be clearly explained to your readers. As a social psychologist who is very familiar with impression management, I found this part of the paper difficult to comprehend. The same is true for unsupportive relationships. It is initially described as between partners but later changed to different types of interpersonal relationships.

Third, the uniqueness of Taiwan (Asian culture) needs to be fleshed out more. We see this developed more in your discussion, but only glossed over in your background. For example, research has shown loneliness to be associated with peer attachment but the relationship between loneliness and cultural values is less clear. Does the concept of collectivism hold water then or can the difference be attributed to Taiwan's geography  (e.g. Durkheim's mechanical vs. organic solidarity). If it is collectivism, then that should be more clearly highlighted in the front end.

Author Response

  1. This is an interesting study on perceived loneliness between two regions of Taiwan. While the methods and analysis are clearly presented, the front end of the paper needs quite a bit of revision. For one, loneliness among young adults is well studied and began to gain traction in the early 90s. Next, the relationship between interpersonal relationship (support) and impression management and loneliness has also been widely researched, particularly in the last 10 years. I would like to see this acknowledged in your review of the literature (e.g. Ponzetti (1990), Wang et al. (2020)) as the initial implication is that the variables used in this study are somewhat novel.

Response:

We have revised both sections of Introduction and Literature Review accordingly.

  1. Next, the variables need to be more clearly operationalized. Impression management in particular needs to be clearly explained to your readers. As a social psychologist who is very familiar with impression management, I found this part of the paper difficult to comprehend. The same is true for unsupportive relationships. It is initially described as between partners but later changed to different types of interpersonal relationships.

Response:

To avoid confusion, we have revised “impression management” to “authentic self-presentation”. Please review the sections of Introduction (pages 1-2, lines 39-40, 50-56), Literature Review (pages 3-4, lines 110-161) and Method 3.3 measurement (pages 6-7, lines 247-253).

  1. Third, the uniqueness of Taiwan (Asian culture) needs to be fleshed out more. We see this developed more in your discussion, but only glossed over in your background. For example, research has shown loneliness to be associated with peer attachment but the relationship between loneliness and cultural values is less clear. Does the concept of collectivism hold water then or can the difference be attributed to Taiwan's geography (e.g. Durkheim's mechanical vs. organic solidarity). If it is collectivism, then that should be more clearly highlighted in the front end.

Response:

We have rewritten the Literature Review section. Please review pages 3-4 (lines 110-196) of the revised document.

Reviewer 3 Report

The manuscript "Urban–Rural Comparison of the Association Between Unsupportive Relationship, Perceived Stress, Impression Management, and Loneliness Among Young Adults in Taiwan " brings an interesting discussion about how unsupportive relationships, stress and impression management influence loneliness in young adults from urban and rural places in Taiwan. Although I found the way the manuscript was conducted, the good chaining of ideas and the writing to be satisfactory, some details were not completely clear:

What knowledge gap does this text fill? What remains to be produced on this? What evidence of this gap is available?

I did not see the sample calculation. Is the sample studied representative of the entire population of the city? This is important for the sample to have the same general characteristics as the study population. The researchers should present the sample size and how it was reached.

Another aspect that can be mentioned for a better understanding of the statistical analysis, in the procedure section, the tests that were performed for each variable studied. Also, what test was used for descriptive statistics.

Author Response

The manuscript "Urban–Rural Comparison of the Association Between Unsupportive Relationship, Perceived Stress, Impression Management, and Loneliness Among Young Adults in Taiwan " brings an interesting discussion about how unsupportive relationships, stress and impression management influence loneliness in young adults from urban and rural places in Taiwan. Although I found the way the manuscript was conducted, the good chaining of ideas and the writing to be satisfactory, some details were not completely clear:

  1. What knowledge gap does this text fill? What remains to be produced on this? What evidence of this gap is available?

Response:

We have added more details and revised in the Introduction part. Please review page 2 (lines 50-56) of the revised document.

  1. I did not see the sample calculation. Is the sample studied representative of the entire population of the city? This is important for the sample to have the same general characteristics as the study population. The researchers should present the sample size and how it was reached.

Response:

We have added more details in Method section. The sample was retrieved from the database of the Taiwan Social Change Survey, which is consistent with the same general characteristics of the Taiwanese population. Please review pages 4-5 (lines 199-215) of the revised document.

Furthermore, we have provided more explanations on the sample of urban/rural young adults. Please review page 5 (lines 217-229) of the revised document.

To avoid statistical error, we kept the sample sizes of two regions balanced; hence, the sample sizes of urban/rural regions might not be representative of the Taiwanese population distribution. We have addressed this problem in the limitation section. Please review page 12 (lines 485-489) of the revised document.

  1. Another aspect that can be mentioned for a better understanding of the statistical analysis, in the procedure section, the tests that were performed for each variable studied. Also, what test was used for descriptive statistics.

Response:

We have written this part. Please review page 7 (lines 261-265) of the revised document.

Round 2

Reviewer 2 Report

The manuscript has significantly improved since I read it last. I appreciate the additions to the literature review and the re-naming of your variables to align with the concepts in the theory. The methods and results are also clearer with the revisions as well.